# Pelvic Sentinel Lymph Node Biopsy for Endometrial Cancer with Multi-Modal Infrared Signal Technology: A Video Article

**DOI:** 10.3390/healthcare12171752

**Published:** 2024-09-03

**Authors:** Federica Perelli, Emanuele Arturo Fera, Marco Giusti, Alberto Mattei, Giuseppe Vizzielli, Martina Arcieri, Gabriele Centini, Errico Zupi, Giovanni Scambia, Anna Franca Cavaliere, Giulia Rovero

**Affiliations:** 1Azienda USL Toscana Centro, Gynecology and Obstetrics Department, Santa Maria Annunziata Hospital, 50012 Florence, Italy; emanuelearturo.fera@uslcentro.toscana.it (E.A.F.); marco1.giusti@uslcentro.toscana.it (M.G.); alberto.mattei@uslcentro.toscana.it (A.M.); giulia.rovero@uslcentro.toscana.it (G.R.); 2Clinic of Obstetrics and Gynecology, “Santa Maria della Misericordia” University Hospital, Azienda Sanitaria Universitaria Friuli Centrale, 33100 Udine, Italy; giuseppe.vizzielli@uniud.it (G.V.); martina.arcieri@asufc.sanita.fvg.it (M.A.); 3Department of Molecular and Developmental Medicine, Obstetrics and Gynecological Clinic, University of Siena, 53100 Siena, Italy; centini.gabriele@gmail.com (G.C.); errico.zupi@unisi.it (E.Z.); 4Division of Gynecologic Oncology, Fondazione Policlinico Universitario A. Gemelli-IRCCS, 00168 Rome, Italy; giovanni.scambia@policlinicogemelli.it; 5Istituto di Clinica Ostetrica e Ginecologica, Università Cattolica del Sacro Cuore, 00168 Rome, Italy; 6Division of Gynecology and Obstetrics, Isola Tiberina Gemelli Hospital, 00186 Rome, Italy; annafranca.cavaliere@fbf-isola.it

**Keywords:** sentinel lymph node, endometrial cancer, infrared signal technology, indocyanine green, gynecologic oncology

## Abstract

This video article summarizes a case study involving the use of pelvic sentinel lymph node (SLN) biopsy for endometrial cancer (EC) staging and treatment utilizing a multi-modal infrared signal technology. This innovative approach combines cervical injection of fluorescent dye indocyanine green (ICG) and near-infrared imaging to enhance SLN detection rates in early-stage EC patients. The study showcases the successful application of advanced technology in improving surgical staging procedures and reducing postoperative morbidity for patients. Multi-modal infrared signal technology consists of different modes of fluorescence imaging used to identify lymph nodes based on near-infrared signals. Each mode serves a specific purpose: overlay image combines white light and near-infrared signals in green, monochromatic visualization shows near-infrared signal in greyscale, and intensity map combines signals in a color scale to differentiate signal intensity. Yellow denotes strong near-infrared signals while blue represents weaker signals. By utilizing a multi-modal approach, surgeons can accurately identify and remove SLN, thus avoiding unnecessary removal of secondary or tertiary echelons.

## 1. Introduction

Endometrial cancer (EC) is the predominant gynecological malignancy in developed countries, with 65,950 new cases reported in 2022 and a 5-year relative survival rate of 81.3% [1].

The standard surgical staging for early-stage EC involves total hysterectomy, bilateral salpingo-oophorectomy, and nodal assessment, including pelvic lymphadenectomy with or without para-aortic lymphadenectomy [2].

Since 2016, sentinel lymph node (SLN) biopsy has emerged as an alternative to systematic lymph node dissection for staging early-stage EC, aiming to reduce post-operative morbidity and long-term complications associated with lymphadenectomy.

Among various tracers and injection sites proposed for SLN mapping, the cervical injection of fluorescent dye indocyanine green (ICG) has demonstrated the highest bilateral pelvic detection rate and is currently recommended [3].

According to the National Comprehensive Cancer Network (NCCN) guidelines, SLN biopsy in EC patients should adhere to the SLN algorithm, which includes retroperitoneal evaluation with excision of any suspicious enlarged nodes irrespective of mapping results, and side-specific pelvic and para-aortic lymphadenectomy in cases of unmapped hemi-pelvis [4,5].

Here, we present a case of a pelvic SLN biopsy for the surgical staging of an early-stage EC patient at our institution, utilizing a multi-modal infrared signal technology.

## 2. Presentation of the Case

A 72-year-old woman, gravida 4, para 2, was diagnosed with endometrioid G1 adenocarcinoma following a hysteroscopic endometrial biopsy for abnormal uterine bleeding (AUB) and was referred to our department in March 2021.

She experienced menarche at 11 years old and reached menopause at 52 years old. Previously, she had breast cancer, and she underwent a right mastectomy with sentinel lymph node biopsy in 2018, followed by adjuvant hormone therapy.

The patient presented with AUB and subsequently underwent hysteroscopy with endometrial biopsy. Pathological evaluation of the biopsy revealed the presence of endometrioid endometrial carcinoma.

An abdomino–pelvic magnetic resonance imaging and a chest–abdomen computerized tomography scan were performed, detecting a 2 cm polypoid mass in the uterine cavity without myometrial invasion, with normal ovaries and no signs of cancer spread, indicating stage IA EC according to the FIGO 2009 classification. The patient underwent laparoscopic surgical staging. After induction of general anesthesia, cervical injection was performed using ICG dye. One 25 mg/vial kit of ICG was diluted in 20 mL of aqueous injectable solution. ICG cervical injection was then performed at 3 o’clock and 9 o’clock positions, injecting 1 mL deeply in the cervix, parallel to the cervical channel, and 1 mL superficially on each side with a 22 gauge needle.

According to a recent Delphi consensus for the standardization of the SLN biopsy surgical technique [6], pelvic lymph node dissection started mobilizing the tissue lateral to the external iliac artery and vein and moving this tissue until the external iliac artery was identified (Figure 1).

The dissection should be performed close to the artery to facilitate resection. After lateral traction of the lymph node with a non-traumatic grasper, it is important to expose the lymphatic trunks and coagulate them to avoid lymphatic spillage and the associated risk of postoperative lymphedema. The landmarks of the pelvic space are the external iliac vessels, the mesoureter with the ureter, the uterine artery, the umbilical artery, the posterior leaf of the broad ligament, and the infundibolo–pelvic ligament.

During pelvic lymph node dissection, a multimodal infrared signal technology was employed to locate a sentinel external iliac lymph node. Different modes of fluorescence imaging were utilized to aid in identifying the lymph node based on near-infrared signals. The available modes are as follows:

(1) The overlay image combines the white light image and the detected near-infrared signal. The near-infrared signal is displayed in green together with the white light signal (Figure 2).

(2) The monochromatic visualization displays the near-infrared signal in greyscale excluding any white light signal (Figure 3). This mode enhances the contrast of the detected near-infrared signal and serves to confirm the uptake of ICG in sentinel lymph nodes.

(3) The intensity map combines the white light image and the detected near-infrared signal. The infrared signal information is displayed in a color scale to differentiate between strong and weak near-infrared signals. The yellowish display indicates stronger signal intensities, while the blue display corresponds to weaker signal intensities (Figure 4).

The optical system used to reveal SLN by the multimodal approach was IMAGE1 S™ Rubina®—4K, 3D and NIR/ICG, by KARL STORZ SE & Co. KG, Tuttlingen, Germany. This advanced technology greatly assisted in the detection of the SLN during the pelvic lymph node dissection procedure (Appendix A).

Following the surgical staging procedure, the patient was discharged two days later. The final pathological examination confirmed a diagnosis of pT1a G1 endometrioid EC, with no evidence of lymph node metastasis in the two pelvic sentinel lymph nodes analyzed, consistent with FIGO staging 2009 IA. The patient was subsequently included in a gynecologic oncological surveillance program, and she is currently free of disease with a disease-free survival and an overall survival of 38 months. The most recent follow-up was performed in June 2024 during which a physical examination and an abdominal and pelvic transvaginal ultrasound were performed, revealing no evidence of recurrence.

## 3. Discussion

The present patient was submitted to the gold standard procedures to study AUB after menopause, hysteroscopy with endometrial biopsy, which revealed endometrioid EC after the pathological evaluation.

Since 2023, the FIGO staging has been modified according to the molecular biology advancements highlighted in the European Society of Gynecological Oncology (ESGO), the European Society for Radiotherapy and Oncology (ESTRO) and the European Society of Pathology (ESP) guidelines published by Concin et al. in 2021 [7,8].

Our patient was referred to hysteroscopy, radiological, and surgical staging in 2021, prior to the routine acquisition in our institution of the clinical assessment through molecular biology of EC according to the new FIGO staging system.

Patients with a diagnosis of pure endometrioid EC and a radiological preoperative staging that showed a disease limited to the uterus should be surgically staged. Minimally invasive surgery (MIS) is the preferred approach when technically feasible, both for the surgical approach (laparoscopy versus laparotomy) and for the surgical technique (SLN identification and removal versus systematic pelvic/aortic lymphadenectomy). SLN mapping is preferred even if excision of suspicious or enlarged lymph nodes in the pelvic or aortic regions is important to exclude nodal metastasis. When the pathological evaluation confirmed a definitive FIGO 2009 IA stage disease (associated with other low-risk factors such as endometrioid histology, low grading G1, no LVSI, and tumor suitable for primary surgery that is completely staged) the patient is suitable for oncological surveillance through exclusive gynecological follow-up.

Total hysterectomy, bilateral salpingo-oophorectomy, and lymph node assessment constitute the gold standard for the surgical staging of apparent uterine confined EC. The lymph node assessment includes evaluation of the regional lymphatic basin and often comprises pelvic lymphadenectomy with or without para-aortic lymphadenectomy at the attending physician’s discretion [2].

Despite the utility of lymph node staging in guiding adjuvant treatment decisions, this approach is not without limitations [9,10]. The procedure is associated with an increased risk of morbidity, including prolonged surgical times, significant blood loss, and heightened rates of complications such as lymphatic damage and lymphedema, lymphocyst formation, and nerve injury [11]. Furthermore, technical difficulties are encountered when performing lymphadenectomy in obese patients, who comprise a substantial proportion of those with EC. To address these limitations, since 2016, the NCCN guidelines have recognized SLN biopsy as a viable alternative to systematic lymphadenectomy for staging apparent uterine-confined EC [4]. SLN biopsy focuses on identifying the first node in the lymphatic drainage basin, effectively allowing for targeted assessment of lymphatic spread while preserving adjacent healthy lymphatic tissue. Consequently, this procedure has shown the potential to significantly reduce the risk of post-operative morbidity and long-term complications, specifically regarding the lower incidence of lymphedema and shorter recovery times. Moreover, it may be accompanied by more intensive pathologic assessment, namely ultrastaging, which enables the detection of low-volume metastases that might be overlooked by standard histological examination [12].

Prospective and retrospective studies have consistently demonstrated that SLN mapping with ultrastaging yields a higher detection rate of lymph node metastasis with low false-negative rates in patients with apparent uterine-confined disease, compared to systematic lymphadenectomy [13]. Furthermore, recent evidence suggests that SLN mapping may also be applied in high-risk histology, such as serous carcinoma, clear cell carcinoma, and carcinosarcoma [14,15].

The integration of ICG and near-infrared imaging technology in SLN mapping has significantly advanced the surgical management of early-stage EC [16]. This innovative approach has not only improved SLN detection rates but has also reduced the total number of lymph nodes removed during staging surgery, potentially decreasing postoperative morbidity for patients [17].

By injecting ICG into the cervix and utilizing near-infrared technology, surgeons can easily identify and highlight the primary SLN [18]. Several trials have demonstrated the high sensitivity and negative predictive value of this method, showcasing its superiority over traditional techniques like isosulfan blue dye [3].

While the use of ICG and near-infrared technology for SLN biopsy has shown promising results, challenges exist in distinguishing true SLN from ‘secondary lymph nodes’. Understanding the potential pitfalls of ICG mapping failures is essential [19]. Surgeons must exercise caution to ensure that the removed SLN is not just adipose tissue or lymphatic vessels [20]. Our case report demonstrates that technological advancements, specifically multi-modal infrared technology, improve imaging capabilities and facilitate the accurate identification of sentinel lymph nodes.

This work emphasizes the significance of a multidisciplinary approach in the treatment of EC, which requires a specialized team with expertise in imaging, surgery, and pathology. Once EC is diagnosed through an endometrial biopsy, it is essential for the patient to be evaluated by radiologists with specialized expertise in gynecologic oncology to ascertain both the local and systemic extent of the malignancy. Subsequently, the gynecologic surgeon will devise a surgical plan based on the preoperative staging of the disease. Following this, a pathologist with proficiency in gynecologic oncology will furnish detailed histopathological findings to the gynecologic oncologist. Factors like age, tumor grade, depth of myometrial invasion, lymph node involvement, and other risk factors are crucial in determining the prognosis for EC patients. This information is critical for determining the necessity of adjuvant therapies and for formulating a comprehensive management plan for oncologic follow-up. It is imperative that a proficient team of gynecologic oncologists is involved in the primary management of all EC patients to ensure optimal clinical outcomes.

While considering the expertise of the medical team managing the EC patient, technological advancements like multi-modal infrared technology allow for the optimization of imaging and facilitate the identification of sentinel lymph nodes.

In conclusion, the multi-modal infrared signal technology used at our institution can greatly facilitate the accurate identification and removal of SLN, avoiding the removal of secondary or tertiary echelons, improving the surgical staging of EC patients.

## Figures and Tables

**Figure 1 healthcare-12-01752-f001:**
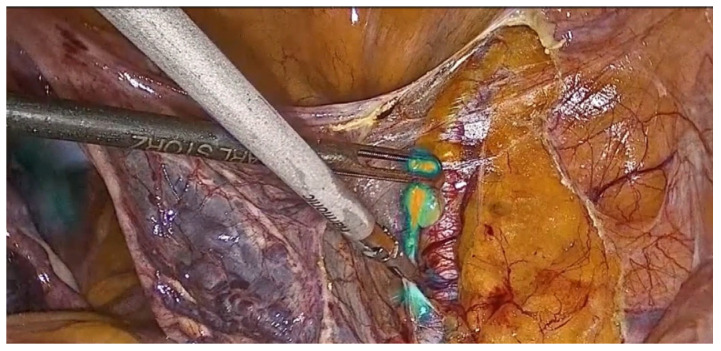
Lateral traction of the lymph node with a non-traumatic grasper to expose the lymphatic trunks and coagulate them to avoid lymphatic spillage and the associated risk of postoperative lymphedema.

**Figure 2 healthcare-12-01752-f002:**
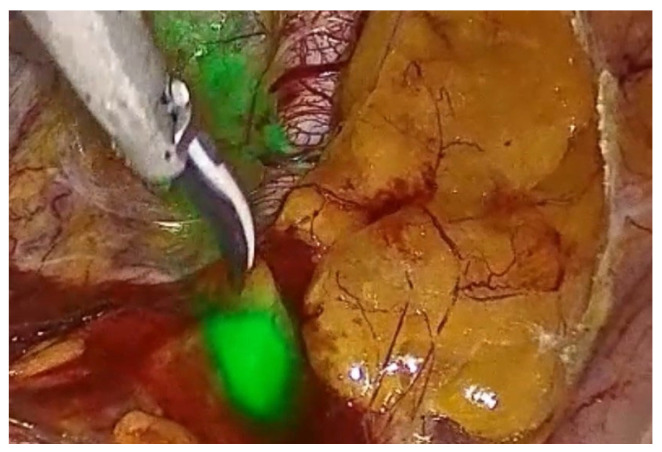
Overlay image visualization of the sentinel lymph node.

**Figure 3 healthcare-12-01752-f003:**
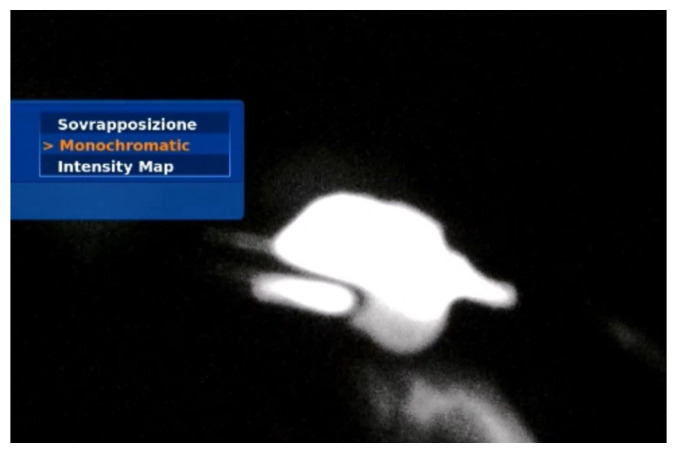
Monochromatic visualization of the sentinel lymph node.

**Figure 4 healthcare-12-01752-f004:**
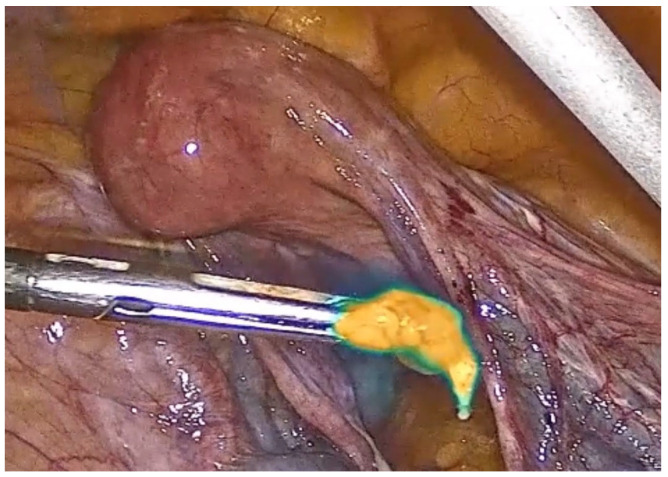
Intensity map visualization of the sentinel lymph node.

## Data Availability

The original contributions presented in the study are included in the article/Appendix A, further inquiries can be directed to the corresponding authors.

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
