# Peer review of "Pelvic Sentinel Lymph Node Biopsy for Endometrial Cancer with Multi-Modal Infrared Signal Technology: A Video Article"

_healthcare, 2024, doi:10.3390/healthcare12171752_

Round 1

Reviewer 1 Report

Comments and Suggestions for Authors

Perelli and colleagues presented an interesting video case report describing the use of multi-modal infrared signal technology for the identification and resection of sentinel lymph nodes in endometrial cancer. The authors clearly presented the methodology and their case report and there are no evident weaknesses. Below are reported some minor comments that will further improve the quality of the manuscript:

1) Lines 61-62: Improve the grammar of the following sentence: “She developed AUB so she was submitted to hysteroscopy with endometrial biopsy demonstrating at the pathological evaluation an endometrioid EC.”. Avoid the use of “so” and use an alternative verb to “submitted to”;

2) Please be more detailed in describing the utility of the monochromatic visualization of the sentinel lymph node;

3) Throughout the manuscript, please use the more appropriate term “near-infrared” instead of “near infrared”;

4) In line 101, please improve the grammar of the following sentence: “The yellowish display represents stronger signals, the blue display weaker signals (Fig.4).”;

5) Improve the grammar. “Since 2016” should be moved to the beginning of the sentence;

6) In the final part of the Discussion section, please briefly mention which professionals are needed for such an approach and the importance of a multidisciplinary approach for the treatment of gynecological tumors (see recent publications on this topic).

Comments on the Quality of English Language

Throughout the manuscript there are some grammar errors. Please carefully check and edit the text.

Author Response

Dear Reviewers and Editors,

Thank you for taking the time to review this article and providing rigorous feedback.

Your constructive reviews and insightful comments have been invaluable in enhancing the quality of the work.

I have carefully reviewed each of your suggested revisions and made the necessary changes.

Below you can find each question raised by the Reviewers, followed by my response, as well as the position in the paper where issue is mentioned.

The entire manuscript has been revised again by a native English speaker, Professor Georgina Porro.

I submitted the revised manuscript highlighting changes made in the text. Page and numbers refer to the revised paper.

REVIEWER #1

Perelli and colleagues presented an interesting video case report describing the use of multi-modal infrared signal technology for the identification and resection of sentinel lymph nodes in endometrial cancer. The authors clearly presented the methodology and their case report and there are no evident weaknesses. Below are reported some minor comments that will further improve the quality of the manuscript:

Comment #1

Lines 61-62: Improve the grammar of the following sentence: “She developed AUB so she was submitted to hysteroscopy with endometrial biopsy demonstrating at the pathological evaluation an endometrioid EC.”. Avoid the use of “so” and use an alternative verb to “submitted to”

  1. a) Author response: We thank the Reviewer for the correction.
  2. b) Location: Page 2; lines 62-64

Comment #2

Please be more detailed in describing the utility of the monochromatic visualization of the sentinel lymph node

  1. a) Author response: We thank the Reviewer for the suggestion. We extended this topic within the “Presentation of the case” section of the revised manuscript, as suggested.
  2. b) Location: Page 3; lines 100-103

Comment #3

Throughout the manuscript, please use the more appropriate term “near-infrared” instead of “near infrared”

  1. a) Author response: We thank the Reviewer for the correction.

Comment #4

In line 101, please improve the grammar of the following sentence: “The yellowish display represents stronger signals, the blue display weaker signals (Fig.4).”

  1. a) Author response: We thank the Reviewer for the suggestion.
  2. b) Location: Page 3; lines 113-115

Comment #5

Improve the grammar. “Since 2016” should be moved to the beginning of the sentence

  1. a) Author response: We thank the Reviewer for the correction.
  2. b) Location: Page 5; lines 168

Comment #6

In the final part of the Discussion section, please briefly mention which professionals are needed for such an approach and the importance of a multidisciplinary approach for the treatment of gynecological tumors (see recent publications on this topic).

  1. a) Author response: We thank the Reviewer for the interesting advice. We discussed this issue in the Discussion section of the revised manuscript, as suggested.
  2. b) Location: Page 6; lines 201-218

Reviewer 2 Report

Comments and Suggestions for Authors

The submitted manuscript-case report "Pelvic sentinel lymphnode biopsy for endometrial cancer with multi-modal infrared signal technology: a video article" describes, illustrates and discusses the use of advanced technology of Indocyanine Green (ICG) fluorescent dye tracing and near-infrared imaging for improving surgical staging and treatment of endometrial cancer (EC). This approach combined cervical injection of ICG and the multi-modal infrared signal technology with different modes of fluorescence imaging to enhance SLN detection rates in early-stage EC patients. The multi-modal infrared signal technology with different modes of fluorescence imaging allowed the successful identification of lymph nodes.

The manuscript carefully describes and illustrates the procedures for the ICG injection through the cervical canal, SLN mapping and identification, and the subsequent surgical manipulations for the SLN removal. The presented manuscript, as well as other clinical studies, convincingly demonstrate that SLN mapping technology performed with ICG and near-infrared imaging significantly advanced the surgical management of early-stage EC by improving SLN detection rates and reducing the total number of lymph nodes removed during surgical staging. Authors conclude that the multi-modal infrared signal technology can facilitate the accurate identification and removal of SLNs, avoiding the removal of secondary or tertiary echelons. They believe that SLN biopsy performed with multi-modal infrared signal technology benefits patients because it potentially reduces postoperative morbidity and may be accompanied by more intensive pathologic assessment (ultrastaging).

The case description is accurate and well illustrated in the Figures and video. In the Discussion section, the authors critically analyze the results of previous clinical trials. The list of references includes the relevant publications.

However, I have the following comments on the manuscript.

1. It is necessary to formulate the novelty of this study, in particular, in comparison with the publication of Frumovitz M et al., Lancet Oncol., 2018. What new facts of this case can be useful to gynecological oncologists that have not previously been announced or confirmed in other works?

In my opinion, the peculiarity of this clinical case study is the presentation of information on successful postoperative consequences after performing pelvic sentinel lymph node (SLN) biopsy using multimodal infrared signal technology (L112: The patient is currently free of disease with a disease-free survival and an overall survival of 38 months...  and pelvic transvaginal ultrasound revealed the absence of any recurrence").

2. It is necessary to specify the manufacturer of the near-infrared fluorescence imaging platform.

3. It is also highly desirable to provide comprehensive information on the histology of the removed sentinel lymph nodes, confirming the presence or absence of cancer cells. This will ensure the thoroughness and reliability of the study.

4. Indicate what anesthesia was administered during the surgical procedure.

5. In the Discussion, it is important to emphasize the absence of side effects in the patient after the SLN biopsy procedure compared to other methods.

Author Response

Dear Reviewers and Editors,

Thank you for taking the time to review this article and providing rigorous feedback.

Your constructive reviews and insightful comments have been invaluable in enhancing the quality of the work.

I have carefully reviewed each of your suggested revisions and made the necessary changes.

Below you can find each question raised by the Reviewers, followed by my response, as well as the position in the paper where issue is mentioned.

The entire manuscript has been revised again by a native English speaker, Professor Georgina Porro.

I submitted the revised manuscript highlighting changes made in the text. Page and numbers refer to the revised paper.

The submitted manuscript-case report "Pelvic sentinel lymphnode biopsy for endometrial cancer with multi-modal infrared signal technology: a video article" describes, illustrates and discusses the use of advanced technology of Indocyanine Green (ICG) fluorescent dye tracing and near-infrared imaging for improving surgical staging and treatment of endometrial cancer (EC). This approach combined cervical injection of ICG and the multi-modal infrared signal technology with different modes of fluorescence imaging to enhance SLN detection rates in early-stage EC patients. The multi-modal infrared signal technology with different modes of fluorescence imaging allowed the successful identification of lymph nodes.

The manuscript carefully describes and illustrates the procedures for the ICG injection through the cervical canal, SLN mapping and identification, and the subsequent surgical manipulations for the SLN removal. The presented manuscript, as well as other clinical studies, convincingly demonstrate that SLN mapping technology performed with ICG and near-infrared imaging significantly advanced the surgical management of early-stage EC by improving SLN detection rates and reducing the total number of lymph nodes removed during surgical staging. Authors conclude that the multi-modal infrared signal technology can facilitate the accurate identification and removal of SLNs, avoiding the removal of secondary or tertiary echelons. They believe that SLN biopsy performed with multi-modal infrared signal technology benefits patients because it potentially reduces postoperative morbidity and may be accompanied by more intensive pathologic assessment (ultrastaging).

The case description is accurate and well illustrated in the Figures and video. In the Discussion section, the authors critically analyze the results of previous clinical trials. The list of references includes the relevant publications.

However, I have the following comments on the manuscript.

Comment #1

It is necessary to formulate the novelty of this study, in particular, in comparison with the publication of Frumovitz M et al., Lancet Oncol., 2018. What new facts of this case can be useful to gynecological oncologists that have not previously been announced or confirmed in other works?

In my opinion, the peculiarity of this clinical case study is the presentation of information on successful postoperative consequences after performing pelvic sentinel lymph node (SLN) biopsy using multimodal infrared signal technology (L112: The patient is currently free of disease with a disease-free survival and an overall survival of 38 months...  and pelvic transvaginal ultrasound revealed the absence of any recurrence").

  1. a) Author response: We thank the Reviewer for the interesting suggestion. We have addressed this issue in the Discussion section of the revised manuscript, as recommended. As demonstrated by Frumovitz et al, Indocyanine Green dye is currently recommended for sentinel lymph node biopsy in endometrial carcinoma due to its superiority in terms of pelvic detection rate compared to other dyes. However, a percentage of cases still experience failure of sentinel lymph node uptake, or the tissue harvested during surgery may not contain nodal tissue. Multimodal infrared technology serves as an intraoperative aid for accurate surgical staging.
  2. b) Location: Page 6; lines 194-218

Comment #2

It is necessary to specify the manufacturer of the near-infrared fluorescence imaging platform.

  1. a) Author response: We thank the Reviewer for the advice. Regrettably, we are unable to disclose the manufacturer of the near-infrared fluorescence imaging platform, as this study is not sponsored.

Comment #3

It is also highly desirable to provide comprehensive information on the histology of the removed sentinel lymph nodes, confirming the presence or absence of cancer cells. This will ensure the thoroughness and reliability of the study.

  1. a) Author response: We thank the Reviewer for the suggestion. We clarified the absence of metastastasis in both sentinel lymph nodes at the final pathology report (consistent with FIGO staging 2009 IA endometrioid endometrial cancer) within the “Presentation of the case” section of the revised manuscript.
  2. b) Location: Page 4; lines 125-127

Comment #4

Indicate what anesthesia was administered during the surgical procedure.

  1. a) Author response: We thank the Reviewer for the suggestion. We specified that the surgical procedure was performed under general anesthesia within the “Presentation of the case” section of the revised manuscript.
  2. b) Location: Page 2; line 72

Comment #5

In the Discussion, it is important to emphasize the absence of side effects in the patient after the SLN biopsy procedure compared to other methods.

  1. a) Author response: We thank the Reviewer for the advice. We discussed this issue in the Discussion section of the revised manuscript, as recommended.
  2. b) Location: Page 5; lines 170-175

Reviewer 3 Report

Comments and Suggestions for Authors

Dear Authors,

I found your article quite interesting, and I appreciate the quality of your presentation.

However, I would like to point out two areas for improvement:

  1. The audio quality of the video is not very clear. Could you please consider enhancing it?

  2. Additionally, please review references 7 and 8 for accuracy.

Author Response

Dear Reviewers and Editors,

Thank you for taking the time to review this article and providing rigorous feedback.

Your constructive reviews and insightful comments have been invaluable in enhancing the quality of the work.

I have carefully reviewed each of your suggested revisions and made the necessary changes.

Below you can find each question raised by the Reviewers, followed by my response, as well as the position in the paper where issue is mentioned.

The entire manuscript has been revised again by a native English speaker, Professor Georgina Porro.

I submitted the revised manuscript highlighting changes made in the text. Page and numbers refer to the revised paper.

I found your article quite interesting, and I appreciate the quality of your presentation.

However, I would like to point out two areas for improvement:

Comment #1

The audio quality of the video is not very clear. Could you please consider enhancing it?

  1. a) Author response: We thank the Reviewer for the suggestion. We have enhanced the audio of the video as you requested.

Comment #2

Additionally, please review references 7 and 8 for accuracy.

  1. a) Author response: We thank the Reviewer for the correction.
  2. b) Location: Page 7; lines 253-260

Round 2

Reviewer 2 Report

Comments and Suggestions for Authors

The revised manuscript was improved by clarifying the issues according to the reviewer's suggestions.

Author Response

Thank you very much for your positive comment.
